# Systemic Metabolic and Volumetric Assessment via Whole-Body [^18^F]FDG-PET/CT: Pancreas Size Predicts Cachexia in Head and Neck Squamous Cell Carcinoma

**DOI:** 10.3390/cancers16193352

**Published:** 2024-09-30

**Authors:** Josef Yu, Clemens Spielvogel, David Haberl, Zewen Jiang, Öykü Özer, Smilla Pusitz, Barbara Geist, Michael Beyerlein, Iustin Tibu, Erdem Yildiz, Sam Augustine Kandathil, Till Buschhorn, Julia Schnöll, Katarina Kumpf, Ying-Ting Chen, Tingting Wu, Zhaoqi Zhang, Stefan Grünert, Marcus Hacker, Chrysoula Vraka

**Affiliations:** 1Department of Biomedical Imaging and Image-Guided Therapy, Division of Nuclear Medicine, Medical University of Vienna, 1090 Vienna, Austria; josef.yu@meduniwien.ac.at (J.Y.); clemens.spielvogel@meduniwien.ac.at (C.S.); david.haberl@meduniwien.ac.at (D.H.); zewen.jiang@meduniwien.ac.at (Z.J.); oeykue.oezer@meduniwien.ac.at (Ö.Ö.); smilla.pusitz@oegk.at (S.P.); barbara.geist@meduniwien.ac.at (B.G.); stefan.gruenert@meduniwien.ac.at (S.G.); marcus.hacker@meduniwien.ac.at (M.H.); 2Christian Doppler Laboratory for Applied Metabolomics, Medical University of Vienna, 1090 Vienna, Austria; 3Department of Otorhinolaryngology, Head and Neck Surgery, Medical University of Vienna, 1090 Vienna, Austria; erdem.yildiz@meduniwien.ac.at (E.Y.); sam.kandathil@meduniwien.ac.at (S.A.K.); till.buschhorn@meduniwien.ac.at (T.B.); julia.schnoell@meduniwien.ac.at (J.S.); 4IT4Science, Medical University of Vienna, 1090 Vienna, Austria; katarina.kumpf@meduniwien.ac.at; 5Teaching Center, Medical University of Vienna, 1090 Vienna, Austria; ying-ting.chen@meduniwien.ac.at; 6Department of Cardiology, Xiangya Hospital Central South University, Changsha 410008, China; wutingting@csu.edu.cn; 7Department of Nuclear Medicine, The Fourth Hospital of Hebei Medical University, Shijiazhuang 050010, China; zhang1979@hebmu.edu.cn

**Keywords:** FDG PET/CT, imaging biomarkers, head and neck cancer, cachexia, artificial intelligence

## Abstract

**Simple Summary:**

Cancer-associated cachexia is a serious complication that can arise in patients with head and neck cancer due to both the disease and its treatments. It causes severe weight loss, muscle and fat depletion, and systemic inflammation, which significantly diminish patients’ quality of life and survival. This study uses advanced machine learning techniques to improve the prediction and understanding of cachexia. By analyzing detailed imaging and clinical data, the research seeks to identify early signs of cachexia, providing insights that could help shape future management strategies. Our findings suggest that specific imaging biomarkers, particularly pancreatic volume, could play a crucial role in predicting cachexia, potentially leading to improved treatment outcomes for affected patients.

**Abstract:**

**Background/Objectives:** Cancer-associated cachexia in head and neck squamous cell carcinoma (HNSCC) is challenging to diagnose due to its complex pathophysiology. This study aimed to identify metabolic biomarkers linked to cachexia and survival in HNSCC patients using [^18^F]FDG-PET/CT imaging and machine learning (ML) techniques. **Methods:** We retrospectively analyzed 253 HNSCC patients from Vienna General Hospital and the MD Anderson Cancer Center. Automated organ segmentation was employed to quantify metabolic and volumetric data from [^18^F]FDG-PET/CT scans across 29 tissues and organs. Patients were categorized into low weight loss (LoWL; grades 0–2) and high weight loss (HiWL; grades 3–4) groups, according to the weight loss grading system (WLGS). Machine learning models, combined with Cox regression, were used to identify survival predictors. Shapley additive explanation (SHAP) analysis was conducted to determine the significance of individual features. **Results:** The HiWL group exhibited increased glucose metabolism in skeletal muscle and adipose tissue (*p* = 0.01), while the LoWL group showed higher lung metabolism. The one-year survival rate was 84.1% in the LoWL group compared to 69.2% in the HiWL group (*p* < 0.01). Pancreatic volume emerged as a key biomarker associated with cachexia, with the ML model achieving an AUC of 0.79 (95% CI: 0.77–0.80) and an accuracy of 0.82 (95% CI: 0.81–0.83). Multivariate Cox regression confirmed pancreatic volume as an independent prognostic factor (HR: 0.66, 95% CI: 0.46–0.95; *p* < 0.05). **Conclusions:** The integration of metabolic and volumetric data provided a strong predictive model, highlighting pancreatic volume as a key imaging biomarker in the metabolic assessment of cachexia in HNSCC. This finding enhances our understanding and may improve prognostic evaluations and therapeutic strategies.

## 1. Introduction

Head and neck squamous cell carcinoma (HNSCC) accounts for approximately 90% of all head and neck cancers and remains a major contributor to the global cancer burden, with over 830,000 new cases and 430,000 deaths annually [1].

The management of this malignancy involves a multidisciplinary team, including ear, nose, and throat (ENT) specialists, oncologists, radiation therapists, and surgeons, among others, who collaborate on treatments such as chemotherapy, radiotherapy, and surgery [2]. Tumor involvement in essential structures for chewing and swallowing, such as the oral cavity, pharynx, and larynx, can severely impair nutritional intake. Post-surgical complications, along with side effects from radiotherapy and chemotherapy, such as mucositis and fibrosis, further contribute to difficulties in eating, potentially leading to malnutrition and cancer-associated cachexia [3,4]. Despite therapeutic advancements, the five-year survival rates for HNSCC vary widely, from 85% in localized cases to 40% in distant metastases [5]. Treatment approaches, including chemotherapy and radiotherapy, can exacerbate this condition by causing complications such as mucositis and fibrotic alterations in the muscles and ligaments of the supraglottic larynx, leading to difficulties in mastication and swallowing [6,7]. Although reduced nutritional uptake may contribute to cachexia, it is insufficient to cause this multi-organ syndrome, which manifests as severe weight loss, altered body composition, and muscle wasting [6,8]. Systemic inflammation also plays a significant role in developing this complex syndrome, with various inflammatory factors, circulating proteins, metabolites, and microRNAs elevated in cachexia [9,10]. The assessment of cachexia has evolved, with the development of various scoring systems and biomarkers for more precise evaluation, including the body mass index (BMI) adjusted weight loss grading system (WLGS) [11], the cachexia score (CASCO) [12], and an inflammation marker (C-reactive protein and albumin concentrations) as part of the Glasgow prognostic score (GPS) [13,14,15]. Additionally, computer tomography (CT) imaging is used to determine the skeletal muscle index (SMI) and diagnose sarcopenia, which is characterized by a loss of muscle mass and function and recognized as a key aspect of cachexia in HNSCC [16]. Extensive research has explored these metrics to identify sarcopenia, a critical criterion for cachexia diagnosis [14,17,18]. However, those measurements have limitations. Inflammatory biomarkers lack specificity [19,20], and imaging-based assessments such as SMI are not widely implemented and offer only a rough estimation of the body composition since they typically utilize two-dimensional data [21].

Positron emission tomography/computed tomography (PET/CT) imaging has become increasingly valuable in the management of HNSCC patients. In clinical practice, [^18^F]FDG-PET/CT is used to assess tumor extent and provide metabolic information. This modality complements CT or magnetic resonance imaging (MRI), the primary imaging modalities for ENT specialists in assessing tumor location and therapy planning [22,23,24]. Furthermore, [^18^F]FDG-PET/CT proves to be an asset in the context of cachexia, as this imaging modality not only indicates tumor metabolism but also tracks increased glucose metabolism, thereby enabling the monitoring of activated immune cells undergoing inflammatory processes [25,26,27,28,29]. Despite advances in total body imaging, the associated highly complex data remain underutilized in clinical practice and, currently, the predominant focus remains on systemic inflammation [3,28,30]. To address these gaps and fully leverage the extensive data provided by using [^18^F]FDG-PET/CT imaging, our study employs machine learning (ML) algorithms, which are particularly well-suited for analyzing complex imaging datasets. By utilizing ML, we can uncover hidden patterns and correlations that may not be immediately apparent through conventional analysis, offering new insights into cachexia in HNSCC. We hypothesize that whole-body [^18^F]FDG-PET/CT imaging, analyzed through ML, can reveal distinct metabolic network patterns associated with varying degrees of weight loss in HNSCC patients, potentially serving as imaging biomarkers for cachexia severity and survival prediction.

## 2. Materials and Methods

### 2.1. Patients

This study retrospectively compiled patient data from two cohorts. The first cohort, from the Vienna General Hospital, included 159 individuals histopathologically diagnosed with HNSCC between 2006 and 2015, all of whom underwent whole-body [^18^F]FDG-PET/CT imaging [31]. Weight information within one year after the initial scan was collected retrospectively using the medical record system, and the patient therapy regimens, including feeding tube use, were examined. To broaden the patient spectrum, we incorporated data from the Head-Neck-CT-Atlas dataset, available through The Cancer Imaging Archive (TCIA), which includes patients from the MD Anderson Cancer Centre. Inclusion criteria for both cohorts included histopathologically confirmed HNSCC, availability of a whole-body [^18^F]FDG-PET/CT scan prior to treatment, completion of treatment course (operation, chemotherapy, radiotherapy, or immunotherapy), and last follow-up. We excluded 88 patients due to the absence of weight follow-up data, 25 for post-processing issues, 7 for incomplete tumor coverage, and 1 for missing clinical characteristics. The study design is depicted in Figure 1.

Initial assessments involved recording patients’ weight during their first pre-treatment [^18^F]FDG-PET/CT scans. These data, along with a subsequent weight measurement, were used to calculate the BMI-adjusted WLGS. The patients were categorized into five stages (WLGS 0 to 4) according to this classification, ranging from no weight loss to severe weight loss (see Appendix B). For our study, we grouped the patients into two cohorts: low weight loss (LoWL; WLGS 0/1/2, representing weight loss below 5%) and high weight loss (HiWL; WLGS 3/4, representing weight loss above 5%). This classification aligns with the criteria established by Fearon et al., where LoWL corresponds to pre-cachexia and cachexia stages, and HiWL corresponds to refractory cachexia [32]. Clinical characteristics recorded included the location and stage of the tumor according to the American Joint Committee on Cancer (AJCC) 8th edition staging system, human papillomavirus (HPV) status, duration until death or latest follow-up, smoking history, usage of a feeding tube, and treatment regimens. Overall survival was defined as the duration from the initial diagnosis or staging scan to the date of death from any cause or the most recent confirmed follow-up.

### 2.2. Image Acquisition and Quantification

In both the Vienna and TCIA cohorts, [^18^F]FDG-PET/CT imaging protocols were rigorously followed. Patients adhered to a fasting period of at least 6 h to ensure blood glucose levels were maintained below 150 mg/dL as in a routine PET/CT scan. After receiving a [^18^F]FDG dose of 148–296 MBq, a 60-min rest period was followed prior to image acquisition. Patients were positioned supine, head first, with arms along the body. For detailed descriptions, refer to Appendix C.

Next, 3D Slicer (version 5.4.0) was utilized for semiautomated lesion segmentation of the primary tumor by a single observer with more than 5 years of clinical experience in diagnostic imaging and supervised by a nuclear medicine physician with more than 10 years of experience. The primary tumor was initially identified using a maximum standardized uptake value (SUV_max) threshold greater than 2.5, with manual adjustments made as necessary to ensure accurate delineation of tumor boundaries. For all subsequent analyses, body weight-corrected standardized uptake values (SUV_bw), referred to hereafter simply as SUVs, were utilized to quantify metabolic activity across various regions of interest (ROIs). Metabolic uptake was quantified using the Multi-Organ Objective SEgmentation tool (MOOSE), which automatically delineated 26 ROIs, including abdominal organs, skeletal muscle, and adipose tissue [33]. Given the study’s focus on body composition, we analyzed fat and muscle tissue both on a whole-body (WB) basis and with a more targeted approach by cropping regions to include only the area between lumbar vertebrae L1 and L5. This allowed us to capture more precise measurements of fat and muscle in key anatomical regions relevant to the study’s objectives. These segmentations were then used to extract SUV values and volumetric information. Representative images and the organs of interest are depicted in Figure 2.

### 2.3. Statistical Analysis

Continuous data are expressed as mean ± standard deviation (SD) or as median and interquartile range (IQR). Categorical variables are presented as numbers and percentages. For patient characteristics, differences in subgroups of numerical variables were calculated using the Mann–Whitney U test, while either Fisher’s exact test or the chi-squared test was used for categorical variables. Weight loss grouping was determined based on specific criteria, categorizing patients into LoWL and HiWL groups. *p* values ≤ 0.05 were considered statistically significant. Statistical analyses were performed using the Python (3.9.5) packages pandas (1.4.2) and scipy (1.9.1).

### 2.4. Network Analysis

Partial correlation coefficients were used to assess inter-organ metabolic correlations, which were then visualized through network analysis to identify key connections and hubs in metabolic interactions. Significant correlations were corrected for multiple comparisons using Bonferroni adjustments. The networks were created and analyzed using the Python (3.9.5) packages networkx (3.3), matplotlib (3.9.2), pandas (1.4.2), and numpy (1.23.1).

### 2.5. Cox Regression Analysis

Kaplan–Meier estimates and Cox regression analyses were used to evaluate the prognostic significance of the individual predictor variables in terms of overall survival (OS). Univariate Cox proportional hazard models were developed first to select significant predictors for inclusion in the multivariate model. Therapy-related parameters resulting from clinical decisions, such as whether a patient received specific treatments, as well as highly redundant features, were excluded. Any missing values were imputed using k-nearest neighbor imputation with the Python package sklearn [34]. The Python packages used for these analyses were lifelines (0.27.8), pandas (1.4.2), numpy (1.23.1), and matplotlib (3.9.2). The simpleNomo (1.0.0) package was used to construct and visualize a nomogram to assess the relative contributions of the covariates in the multivariate analysis.

### 2.6. Machine Learning

Two binary machine learning classifiers were constructed and validated using Monte-Carlo (MC) cross-validation to predict binarized cachexia-related status as defined by the WLGS. The initial classifier was designed to differentiate between LoWL and HiWL. The second classifier extended this approach by incorporating a composite endpoint, which included both HiWL status and overall survival (OS) of less than one year post-scan, thereby classifying patients as cachectic if they met both criteria and non-cachectic in all other cases. To ensure a robust performance estimate, MC cross-validation was performed with 100 folds and an 80 to 20 split ratio between training (*n* = 202) and test (*n* = 51) sets. Data pre-processing included z-score feature standardization, feature selection based on minimum redundancy maximum relevance (mRMR) [35], k-nearest neighbor imputation [34], and class balancing using the synthetic minority oversampling technique (SMOTE) [36]. All pre-processing steps were strictly performed on the training set and afterward applied to the test set to avoid any data leakage and resulting overfitting. For classification, a random forest algorithm was employed. Hyperparameters were optimized using random search through a defined parameter grid. To ensure clinical applicability, classification performance was assessed using key performance metrics [37,38], including accuracy (ACC), sensitivity (SNS), specificity (SPC), positive predictive value (PPV), negative predictive value (NPV), balanced accuracy (BACC), and area under the curve (AUC), along with Shapley additive explanations (SHAPs) [39]. Unsupervised dimensionality reduction was performed using t-distributed stochastic neighbor embedding (t-SNE) [37,38]. For the analysis and visualization, the Python (3.9.5) packages pandas (1.4.2), scikit-learn (1.1.0), imbalanced-learn (0.8.0), shap (0.40.0), mrmr-selection (0.2.5), seaborn (0.11.2), and plotly (5.8.0) were used.

## 3. Results

### 3.1. Clinical Characteristics

In this study, we retrospectively included 253 patients, comprising 114 from the Vienna cohort and 139 from the TCIA database [40]. The median age was 58.5 years (IQR: 57.4–60.0 years) and the average weight was 81.2 ± 20.58 kg. The median follow-up was 42.50 months. Overall survival (OS) rates were 75.49% at 1 year and 40.32% at 5 years. Most patients were male (78.26%) and had a BMI of 25 or higher (62.06%). Tumor origin was predominantly from the oropharynx (67.98%), followed by the larynx (11.07%) and the oral cavity (9.09%). Most patients (69.17%) were at stage IVa, indicating locally advanced tumors with regional spread to adjacent tissues or lymph nodes but without distant metastasis, reflecting a more advanced stage of cancer. A feeding tube was administered in 51.78% of patients, and 69.17% had a positive smoking history. HPV status was positive in 13.83% of patients, negative in 28.85%, and not reported in 57.31%. The detailed characteristics of the study population are summarized in Table 1.

### 3.2. Metabolic and Clinical Profiles by Weight Loss

Significant disparities between the LoWL and HiWL groups were observed in both SUV values and organ volumes, highlighting the distinct metabolic profiles associated with varying degrees of weight loss. The HiWL group showed elevated SUV values in intramuscular adipose tissue (IMAT), subcutaneous adipose tissue (SAT), skeletal muscle (SKM), visceral adipose tissue (VAT), and adrenal glands, while the lungs had significantly lower SUV values. Body composition-related tissue volumes, including IMAT, SAT, VAT, and thoracic adipose tissue (TAT), were notably reduced in the HiWL group, along with reductions in the pancreas, spleen, and gastrointestinal (GI) tract. The pancreas volume reduction was one of the most significant organ changes in the HiWL group compared to the LoWL group (71.51 ± 24.69 vs. 81.34 ± 22.22, *p* < 0.01). Clinically, the 1-year survival rate was lower in the HiWL group (69.18% vs. 84.11%, *p* < 0.01), and a higher prevalence of smoking (76.71% vs. 58.88%, *p* = 0.03). Additionally, HPV status was only available for a subset of patients, and feeding tube use was more commonly associated with the HiWL group (57.53% vs. 43.93%, *p* = 0.04, Appendix A).

### 3.3. HiWL Is Associated with Increased Metabolic Inter-Organ Connectivity

In the LoWL group, five positive correlations were identified, notably between SKM and VAT (R = 0.461, *p* = 0.026) and between IMAT and VAT (R = 0.401, *p* = 0.026). In contrast, the HiWL group displayed a more intricate network, consisting of nine positive correlations and one negative correlation, indicating a higher degree of inter-organ connectivity associated with greater weight loss. Notable correlations in the HiWL group included SKM and VAT (R = 0.376, *p* = 0.011) and SAT and VAT (R = 0.519, *p* < 0.001). Additionally, a negative correlation was observed between the GI tract and SAT (R = −0.336, *p* = 0.029). Organs such as the spleen, GI tract, and adrenal glands showed stable connectivity patterns with minimal changes, while the lungs remained unconnected to other organs in both groups (Figure 3).

### 3.4. Metabolic and Volumetric Features as Key Predictors of Survival in HNSCC

Survival analysis yielded 26 covariates as significant predictors of survival outcomes in univariate Cox proportional hazards models. These significant variables were subsequently included in a multivariate analysis to adjust for potential confounders. Survival analysis within each weight loss group (LoWL and HiWL) was conducted, revealing that feeding tube use had differing associations with survival outcomes. In the LoWL group, feeding tube use showed a non-significant trend toward improved survival (HR: 0.872, 95% CI: 0.497–1.531, *p* = 0.634), whereas in the HiWL group, it was associated with a non-significant trend toward increased risk of death (HR: 1.218, 95% CI: 0.795–1.864, *p* = 0.365). In the multivariate analysis, tumor volume (HR: 1.016, 95% CI: 1.009–1.023, *p* < 0.001), pancreas SUV (HR: 3.033, 95% CI: 1.003–9.174, *p* = 0.049), pancreas volume (HR: 0.984, 95% CI: 0.974–0.994, *p* = 0.002), SKM SUV (HR: 18.211, 95% CI: 1.116–297.138, *p* = 0.042), and SKM Volume (HR: 1.0, 95% CI: 0.999–1.0, *p* = 0.039) were identified as significant independent predictors of survival. A nomogram illustrating the influence of these covariates on survival probability was constructed (Figure 4, Appendix A).

### 3.5. Machine Learning Model for Cachexia and Survival

First, the ML model demonstrated limited accuracy in predicting HiWL alone, with an AUC of 59.68% (95% CI: 0.60–0.61). To determine the impact of specific input features, multiple iterations were conducted, separately evaluating metabolic and volumetric parameters. These iterations revealed minimal variation in metrics, indicating that both parameter types independently contributed substantial predictive value. However, integrating both metabolic and volumetric parameters improved the model’s performance, particularly in predicting the composite endpoint of HiWL and 1-year OS, achieving an AUC of 76.07% (95% CI: 0.75–0.76), with an ACC of 80.98%, an SNS of 51.56%, and an SPC of 87.29%. Subsequently, we assessed the model’s performance across different cohorts. For the Vienna site, the model’s AUC was 75.77% (95% CI: 0.74–0.76). However, for the TCIA site, the model demonstrated a lower AUC of 53.33% (95% CI: 0.51–0.55), reflecting a substantial variation in predictive performance between the cohorts (Appendix A). SHAP analysis, applied to the composite endpoint model that integrated both metabolic and volumetric features across the combined dataset, further emphasized the importance of volume-based imaging parameters such as the volume of the pancreas, right ventricle, and body compartments in enhancing the model’s predictive performance. Of the 16 most important features identified, 11 were volume-based and 4 were SUV-based (see Figure 5).

## 4. Discussion

In this study, we examined the diagnostic potential of 29 tissues and organs together with 12 clinical parameters to detect cachexia in 253 patients with HNSCC. Our results reveal significant metabolic differences between patients in LoWL and HiWL groups, indicating that cachexia involves systemic metabolic alterations beyond muscle wasting and systemic inflammation. While previous research has focused on clinical measures such as modified WLGS, grip strength, and gait speed, our study employs whole-body [^18^F]FDG-PET/CT imaging and ML to probe the metabolic disruptions preceding the severe weight loss currently used to diagnose cachexia [41,42].

Head and neck cancers, particularly oropharyngeal cancer, are frequently associated with HPV infection [43]. In our cohort, oropharyngeal cancer accounted for the highest number of primary tumors, with 42.68% of patients having known HPV status. Our findings are consistent with existing literature that identifies smoking and male sex predominance as significant correlates of cachexia development in this population [44]. Our comprehensive approach, integrating whole-body PET/CT data and clinical characteristics, offers a more detailed diagnostic framework for cachexia.

Recent advancements in the understanding of cachexia have emphasized its systemic nature, characterized by widespread metabolic reorganization [45]. Our network analysis of inter-organ metabolic relationships further supports this, revealing critical inter-organ metabolic relationships. While the overall network structures in the LoWL and HiWL groups appeared similar, partial correlation analysis revealed critical differences that suggest biological alterations linked to severe weight loss. The HiWL group demonstrated an increased number of metabolic connections, indicating a potential compensatory mechanism where glucose utilization becomes more interdependent across various organs in response to altered metabolic demands. This finding aligns with recent studies that describe cachexia as a syndrome of complex metabolic reorganization, involving multiple organs to maintain energy balance in the face of severe illness [46].

In another study, network analysis was used to investigate metabolic abnormalities across various diseases with a limited number of patients [47]. In contrast, our approach focuses on a single disease with a larger patient cohort, offering a unique perspective on how metabolic networks function specifically in cachectic (HiWL) HNSCC patients. Further research on other diseases is needed to determine whether these network alterations are unique to HNSCC or if they represent a common feature of cachexia across different conditions.

From a physiological perspective, our network analysis revealed distinct alterations in metabolic connectivity between the LoWL and HiWL groups, offering insights into the systemic effects of cachexia. For instance, in the LoWL group, the liver maintained its connection with the aorta, reflecting its role in systemic circulation. However, this connection was lost in the HiWL group, with the aorta instead linking to the spleen and pancreas, suggesting a shift in blood flow to prioritize immune defense and glucose regulation, which are key features of the cachectic state. This shift may be driven by molecular signals such as pro-inflammatory cytokines (e.g., IL-6, TNF-α), hormonal imbalances, and dysregulation of key metabolic pathways, including the mTOR and AMPK signaling pathways [48]. mTOR activation, often suppressed in cachexia, is crucial for anabolic processes and muscle growth, while AMPK activation in response to energy stress promotes catabolic processes, exacerbating tissue breakdown [49]. These molecular shifts, coupled with altered insulin and cortisol signaling, may contribute to the prioritization of immune defense and glucose regulation across organs like the spleen and pancreas, key features of the cachectic state [50]. Similar shifts in metabolic connectivity, driven by these hormonal and molecular signals, have been documented in other studies, further underscoring the multifaceted nature of cachexia [51].

The multivariate Cox regression analysis revealed that both the tumor lesion volume and the metabolism and volume of other organs correlate significantly with overall survival. Interestingly, subcutaneous fat, which usually shows low [^18^F]FDG uptake, exhibited higher uptake in cachectic patients, suggesting disrupted fat metabolism and the presence of chronic inflammation. Elevated glucose utilization in adipose tissue could be indicative of immune cell activity, reflecting underlying inflammation that may not be captured by markers like CRP or IL-6, which were unavailable in this cohort [52]. These changes in body composition may play a critical role in disease progression. Additionally, differences in body composition between sexes could contribute to the observed variation in cachexia prevalence in our study [53].

A novel finding In our study is the role of reduced pancreatic volume as a predictor of weight loss and survival. This observation suggests that pancreatic dysfunction, likely due to tissue atrophy, plays a more critical role in the progression of cachexia than previously recognized [54]. Pancreatic fat has been implicated in metabolic syndrome and insulin resistance [55]; however, this is the first study, to our knowledge, to report that reduced pancreatic volume is directly predictive of both weight loss and survival in cachectic patients. The volume reduction may impair endocrine and exocrine functions, disrupting glucose metabolism and digestive processes, both of which are commonly observed in cachexia. While an increase in pancreatic volume may suggest edematous changes, disease-related tissue alterations must also be considered. Identifying pancreatic volume as a predictor of cachexia underscores the underappreciated role of pancreatic dysfunction in this condition. Uncovering the yet unknown pathways underlying these changes is important for developing accurate biomarkers and could also lead to new prevention strategies for cachexia.

In our study, we demonstrated that in direct weight loss comparisons between patient groups, the L1–L5 region showed more significant differences than whole-body analyses. This finding highlights the importance of the L1–L5 region for assessing body composition, as it reduces the impact of variability caused by different scanners, patient anatomy, and scanning protocols, which can lead to differences in anatomical ROIs. Accordingly, other studies have shown that this approach can be effectively used to determine overall survival, reinforcing the value of focusing on the L1–L5 region for robust prognostic assessments [56].

The importance of maintaining nutritional status in HNSCC patients is particularly evident in the context of early cachexia. Our findings from the network analysis highlight the increased inter-organ connectivity in the HiWL group, especially between SKM, VAT, and SAT, suggesting that as weight loss progresses, metabolic interactions between these tissues become increasingly complex and indicative of the systemic nature of cachexia. Consistent with current knowledge, our data reveal significant alterations in glucose metabolism within the pancreas and GI tract, aligning with studies that have shown changes in gut microbiota and gut barrier function during cachexia [57,58,59]. These alterations likely reflect a broader metabolic reorganization towards wasting, where mechanisms such as insulin resistance, mitochondrial dysfunction, and the activation of the ubiquitin-proteasome pathway contribute to muscle and fat loss. Additionally, metabolic uncoupling in white adipose tissue, driven by uncoupling proteins, leads to inefficient energy extraction even with nutritional support, exacerbating the cachectic state [60,61]. Despite these interventions, the body may fail to adequately extract and utilize energy, furthering weight loss [62,63]. These findings emphasize the importance of managing systemic effects to preserve the nutritional status and well-being of HNSCC patients. Despite its common use, enteral nutritional support does not significantly affect survival outcomes, highlighting the need for broader therapeutic strategies to address the complex metabolic changes in cachexia [64].

Our approach revealed distinct metabolic patterns in various organs, correlating with the severity of cachexia and clinical outcomes. Integrating state-of-the-art ML-based segmentation techniques in PET/CT data analysis is becoming increasingly important in clinical practice. While it remains to be seen whether ML will directly transform patient outcomes, its application in routine workflows is likely to benefit clinicians by capturing and analyzing key elements of PET/CT imaging that are often underused or overlooked. Such techniques could not only enhance the accuracy of cachexia diagnosis and monitoring but also provide comprehensive insights that could significantly improve disease management. This capability is crucial, as the patterns we observed confirm that cachexia in HNSCC is not merely a symptom of the primary disease but a complex, systemic syndrome involving multiple organs.

In our comparison of the clinical parameters between the LoWL and HiWL groups, we observed that one-year survival was significantly lower in the HiWL group, with higher rates of feeding tube use and smoking. Despite the increased dependence on feeding tubes in the HiWL group, our survival analysis revealed that feeding tube usage did not significantly impact survival outcomes (Appendix A), underscoring the complexity of cachexia in HNSCC. Furthermore, our study highlights the importance of addressing treatment-induced effects such as xerostomia, dysphagia, and dysgeusia, which are often exacerbated by chemotherapy and radiotherapy. While these conditions significantly reduce food intake and contribute to weight loss, our analysis focused on tube feeding as an intervention and found no direct correlation between tube feeding and the severity of cachexia. This suggests that managing these side effects is crucial for maintaining patient quality of life; however, they do not necessarily drive the progression of cachexia. The interplay between systemic metabolic disruptions and the impact of treatment on nutritional intake requires comprehensive strategies that address both metabolic and nutritional challenges in managing cachexia in HNSCC patients.

Our findings underscore the potential of PET/CT imaging combined with advanced machine learning (ML) techniques for diagnosing and characterizing cachexia. Initially, our ML model, using the WLGS stage as the predictive target, showed limited accuracy in predicting HiWL alone, with an AUC of 59.68%, indicating that weight loss alone may not fully capture the complexity of cachexia. Adjusting the model to predict a composite endpoint of HiWL and 1-year overall survival significantly improved performance, achieving an AUC of 76.07%.

Interestingly, when comparing model metrics across different cohorts, the AUC varied significantly. At the Vienna site, the model achieved an AUC of 75.77%, while at the TCIA site, the AUC dropped to 53.33%. This stark difference underscores the challenges of generalizing predictive models across diverse patient populations, highlighting the need for further refinement and cohort-specific validation (Appendix A). Key predictive factors identified in the SHAP analysis included BMI, pancreatic volume, and FDG uptake values of the lungs and adrenal glands. In another study, similar organs were identified as key predictors of cachexia, with a particular emphasis on the L3 body composition region. However, that study did not incorporate a composite endpoint, which may have limited its model’s performance [65]. This comparison emphasizes the value of a multi-dimensional approach, integrating both metabolic and volumetric parameters, to enhance the predictive accuracy of cachexia models.

The tremendous potential of ML-powered predictive models in clinical practice is evidenced by the growing number of approved medical devices utilizing artificial intelligence (AI) and ML technologies. As of August 2024, there are more than 950 AI and ML-enabled medical devices approved by the FDA alone. The overwhelming majority of these devices are established in the field of medical imaging with 723/950 (76%) [66]. A reliable and early diagnosis of cachexia with such a tool could support the clinical development of novel therapeutic interventions for this devastating condition [65]. Early detection and patient stratification could improve the effectiveness of existing treatments and open the door to more personalized therapeutic strategies. Furthermore, several new pharmacological approaches, such as GDF-15 antibodies, selective androgen receptor modulators (SARMs), and ghrelin receptor agonists like anamorelin, are being explored to target muscle wasting and enhance patient outcomes [67,68,69].

Our study has several limitations that should be considered. Despite efforts to ensure comparability between the Vienna and TCIA cohorts, potential site-specific biases might have influenced the findings. This underscores the need for more standardized cohorts and methodologies in future research. The use of WLGS for cachexia classification, while practical, may not fully capture the complexity of weight loss in cachexia. Incorporating dynamic weight-tracking methods could enhance staging accuracy in subsequent studies. The absence of external validation, despite combining two cohorts and reporting the AUC of our ML model, highlights the importance of independent validation to confirm our conclusions. Additionally, conducting longitudinal PET analyses could provide better insights into causal relationships and early markers of cachexia. This could be further supported by larger cohorts if longitudinal data remain unavailable.

Moreover, the lack of a precise and measurable definition of cachexia in our study, particularly given the absence of laboratory data from the TCIA cohort, limits the robustness of our findings. The predominance of patients with stage IVa in our cohort may restrict the generalizability of the results. Manual tumor delineation and organ segmentation introduce potential errors, and the fact that the patients were not treatment-naïve, along with some missing data such as HPV status, complicates the interpretation of our findings. Finally, differences in the level of characterization between the TCIA and Vienna cohorts could have affected the consistency of our results, emphasizing the importance of using well-characterized cohorts in future research.

## 5. Conclusions

Our research provides novel insights into the systemic nature of cachexia in HNSCC patients, highlighting the potential of [^18^F]FDG-PET/CT imaging in understanding and managing this complex syndrome. The integration of imaging techniques and ML models offers a promising framework for future investigations to predict patient outcomes and potentially lead to better patient management and treatment strategies. Our study emphasizes the potential of a multi-organ approach for understanding and managing cachexia in HNSCC patients. The significant role of pancreatic volume as a biomarker should be further explored in clinical settings, potentially leading to earlier interventions that could improve patient outcomes. Future research should consider longitudinal studies to validate these findings and refine cachexia staging systems.

## Figures and Tables

**Figure 1 cancers-16-03352-f001:**
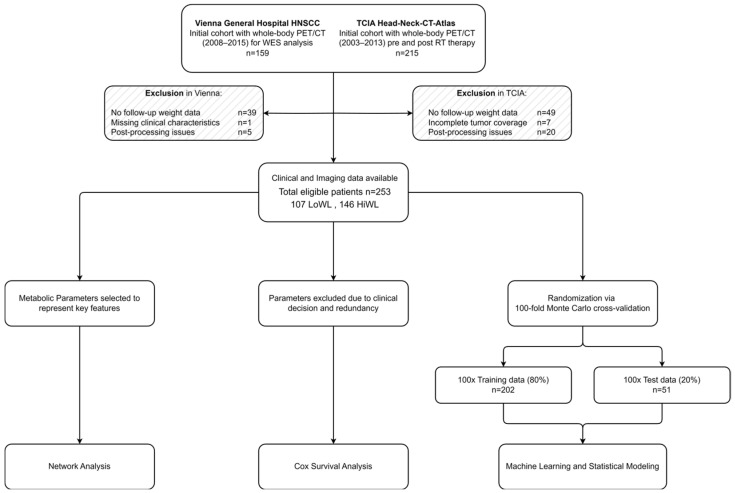
Study design and patient selection. The flowchart depicts the patient selection process and the overall study design. Abbreviations: HNSCC, head and neck squamous cell carcinoma; WES, whole-exome sequencing; RT, radiotherapy; TCIA, the cancer imaging archive.

**Figure 2 cancers-16-03352-f002:**
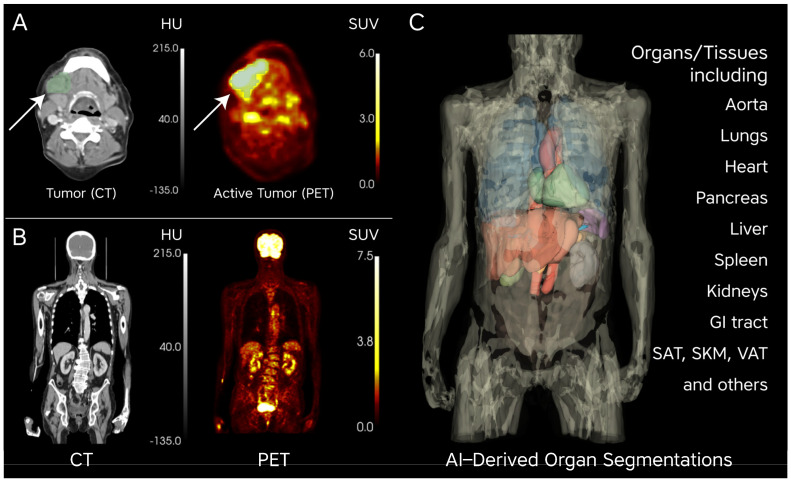
Tumor delineation and metabolic activity visualization: (**A**) Tumor delineation (green) at the 3rd cervical vertebra level. The CT image (**left**, grayscale) shows the anatomical structure, while the PET image (**right**) displays the tumor’s metabolic activity, represented by elevated [^18^F]FDG uptake in standardized uptake value (SUV) units. Warmer colors (yellow to red) indicate higher FDG uptake, suggesting increased metabolic activity in the tumor. (**B**) Coronal view combining CT (**left**) and PET (**right**) images, illustrating both anatomical structure and metabolic function. Warmer colors on the PET image indicate glucose-active organs, such as the brain and bladder, on the SUV scale (0.0–7.5). (**C**) AI-derived organ segmentations in a 3D anatomical model. Each color represents a distinct organ, including the aorta, lungs, heart, pancreas, liver, spleen, kidneys, GI tract, SAT, SKM, VAT, and others. This segmentation allows for SUV extraction and quantification of metabolic activity from [^18^F]FDG PET/CT imaging. Abbreviations: SUV, standardized uptake value; FDG, fluorodeoxyglucose; SAT, subcutaneous adipose tissue; SKM, skeletal muscle; VAT, visceral adipose tissue; GI Tract, gastrointestinal tract; AI, artificial intelligence.

**Figure 3 cancers-16-03352-f003:**
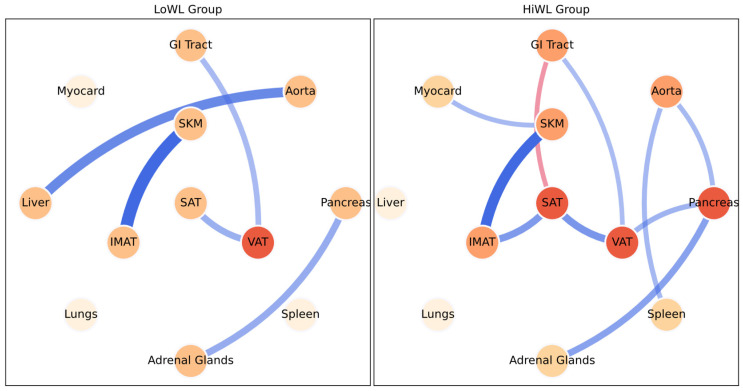
Network analysis of metabolic interactions in LoWL and HiWL groups. Circles represent organs and tissues (nodes), while connections are shown as edges. The color intensity of nodes indicates the degree of connectivity, and the color intensity of edges reflects the strength of the correlation. Blue edges denote positive correlations, red edges denote negative correlations, and thicker edges represent stronger correlations. Abbreviations: SKM, skeletal muscle; IMAT, intramuscular adipose tissue; SAT, subcutaneous adipose tissue; VAT, visceral adipose tissue; GI tract, gastrointestinal tract.

**Figure 4 cancers-16-03352-f004:**
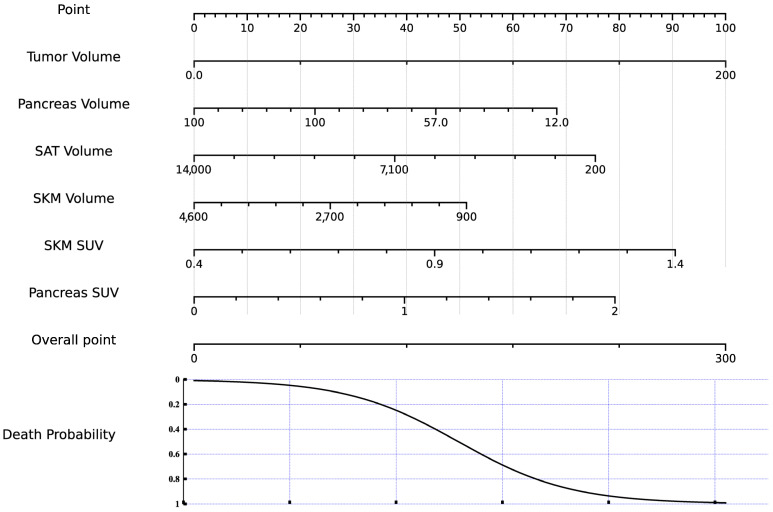
Nomogram for survival prediction. This nomogram assigns points based on imaging parameters from multivariate analysis, which are summed to estimate the probability of death. Abbreviations: SUV, standardized uptake value; SKM, skeletal muscle; SAT, subcutaneous adipose tissue.

**Figure 5 cancers-16-03352-f005:**
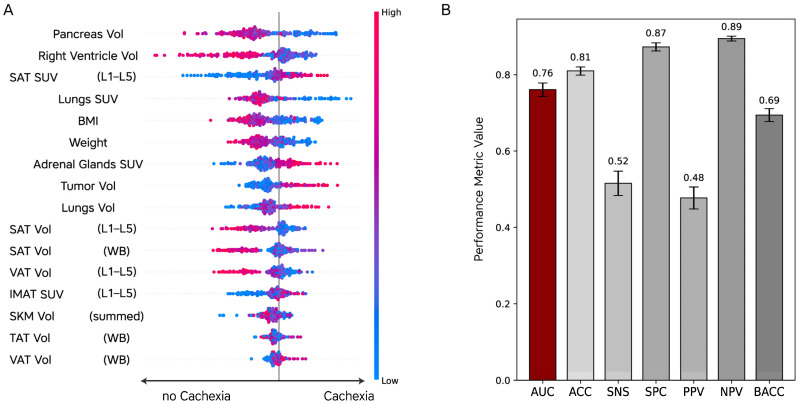
SHAP analysis and model performance (HiWL plus death within 1 year): (**A**) SHAP plot illustrating the influence of various features on the prediction of the composite endpoint, including HiWL and death within one year (cachexia). Feature contributions are represented with a color gradient, where blue indicates lower feature values and red indicates higher feature values; (**B**) bar plots displaying the model’s overall performance metrics. Abbreviations: Vol, volume; SAT, subcutaneous adipose tissue; SAT SUV, subcutaneous adipose tissue standardized uptake value; L1–L5, lumbar vertebra section L1–L5; BMI, body mass index; SAT Vol, subcutaneous adipose tissue volume; VAT Vol, visceral adipose tissue volume; IMAT SUV, intramuscular adipose tissue standardized uptake value; SKM Vol, skeletal muscle volume; WB, whole-body; ACC, accuracy; SNS, sensitivity; SPC, specificity; PPV, positive predictive value; NPV, negative predictive value; BACC, balanced accuracy; AUC, area under the curve.

**Table 1 cancers-16-03352-t001:** Clinical characteristics and treatment regimens across sites.

		Vienna (*N* = 114)		TCIA (*N* = 139)		Total (*N* = 253)	
Age ± SD (years)		60.0 ± 11.55		57.4 ± 9.37		58.5 ± 10.57	
Weight ± SD (kg)		73.1 ± 20.55		87.9 ± 18.70		81.2 ± 20.58	
Follow-up (months)		15.1 ± 36.88		65.0 ± 29.74		42.5 ± 35.65	
Overall survival (OS)							
	1-year OS alive	61	(53.51%)	125	(89.93%)	191	(75.49%)
	5-year OS alive	23	(20.18%)	79	(56.83%)	102	(40.32%)
Sex							
	Male	81	(71.05%)	117	(84.17%)	198	(78.26%)
Body mass index (kg/m^2^)							
	BMI ≥ 25	45	(39.47%)	102	(73.38%)	157	(62.06%)
	BMI < 25	69	(60.53%)	37	(26.62%)	96	(37.94%)
Weight loss grading system (WLGS)							
	WLGS 0	5	(4.39%)	9	(6.47%)	14	(5.53%)
	WLGS 1	19	(16.67%)	27	(19.42%)	46	(18.18%)
	WLGS 2	10	(8.77%)	37	(26.62%)	47	(18.58%)
	WLGS 3	31	(27.19%)	54	(38.85%)	85	(33.60%)
	WLGS 4	49	(42.98%)	12	(8.63%)	61	(24.11%)
Tumor origin							
	Oropharynx	72	(63.16%)	100	(71.94%)	172	(67.98%)
	Larynx	12	(10.53%)	16	(11.51%)	28	(11.07%)
	Oral Cavity	17	(14.91%)	6	(4.32%)	23	(9.09%)
	Hypopharynx	13	(11.40%)	8	(5.76%)	21	(8.30%)
	Nasopharynx	0	(0.00%)	5	(3.60%)	5	(1.98%)
	Cancer unknown primary	0	(0.00%)	4	(2.88%)	4	(1.58%)
Clinical staging							
	I	6	(5.26%)	2	(1.44%)	8	(3.16%)
	II	11	(9.65%)	2	(1.44%)	13	(5.14%)
	III	7	(6.14%)	20	(14.39%)	27	(10.67%)
	IVa	71	(62.28%)	104	(74.82%)	175	(69.17%)
	IVb	9	(7.89%)	11	(7.91%)	20	(7.91%)
	IVc	10	(8.77%)	0	(0.00%)	10	(3.95%)
HPV status							
	Negative	62	(54.39%)	11	(7.91%)	73	(28.85%)
	Positive	11	(9.65%)	24	(17.27%)	35	(13.83%)
	Not reported	41	(35.96%)	104	(74.82%)	145	(57.31%)
Feeding tube							
	Yes	56	(49.12%)	75	(54.74%)	131	(51.78%)
Smoking history							
	Yes	86	(76.79%)	89	(64.03%)	175	(69.17%)
Therapy regimen							
	Surgery	55	(48.25%)	41	(29.50%)	96	(37.94%)
	Neoadjuvant	14	(12.28%)	50	(35.97%)	64	(25.30%)
	Radiotherapy	94	(82.46%)	139	(100.00%)	233	(92.09%)
	Chemotherapy	74	(64.91%)	112	(80.58%)	186	(73.52%)

Clinical characteristics and treatment regimens of 253 patients, represented by absolute numbers and percentages (%). Key variables include age, weight, survival rates, BMI, WLGS, tumor origin, clinical staging, HPV status, feeding tube use, smoking history, and therapy approaches.

## Data Availability

The data supporting the reported results are available upon reasonable request to the corresponding author. Direct access to the data is limited as per IRB guidelines. Data from TCIA is available at https://doi.org/10.7937/K9/TCIA.2017.umz8dv6s.

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
