# Peer review of "Systemic Metabolic and Volumetric Assessment via Whole-Body [18F]FDG-PET/CT: Pancreas Size Predicts Cachexia in Head and Neck Squamous Cell Carcinoma"

_cancers, 2024, doi:10.3390/cancers16193352_

Round 1

Reviewer 1 Report

Comments and Suggestions for Authors

The manuscript titled,” Characterization of Glucose Metabolism as a Biomarker for Cancer associated Cachexia: Whole-Body [¹⁸F]FDG-PET/CT Imaging in Head and Neck Squamous Cell Carcinoma” is very interesting and well written. It focuses on the imaging techniques and scans on oral cancer patients and identify the role of pancreatic volume as a crucial marker for cachexia in affected patients.

My suggestions are as follows:

1.       It would be best to highlight “Pancreatic volume” as a predictor of cachexia in the title.

2.       In table 1, can the author include the available Pancreatic volume data?

3.       Fig5B include Y axis label.

4.       Although it can be indicated that “Pancreatic volume” can be a predictor of cachexia but it is not confirmed. How will the authors clarify that?

5.       Some of the statements are not “totally” correct. Eg. “This malignancy, primarily managed by Ear, Nose, and Throat (ENT) specialists, affects critical areas such as the oral cavity, pharynx, and larynx—regions essential for nutritional intake.” There is a significant contribution from the oncologists, radiation dosimetrists etc…so the authors should edit that sentence. Moreover, I could not see such thing in their cited ref #2.

6.       Again, while reporting the distribution the authors have cited a reference dating back to 2018. They should cite recent papers.

Author Response

The manuscript titled, ”Characterization of Glucose Metabolism as a Biomarker for Cancer associated Cachexia: Whole-Body [¹⁸F]FDG-PET/CT Imaging in Head and Neck Squamous Cell Carcinoma” is very interesting and well written. It focuses on the imaging techniques and scans on oral cancer patients and identify the role of pancreatic volume as a crucial marker for cachexia in affected patients.

My suggestions are as follows:

  1. It would be best to highlight “Pancreatic volume” as a predictor of cachexia in the title.

We agree with your suggestion to highlight "Pancreatic volume" in the title. We have updated the title to: "Systemic Metabolic and Volumetric Assessment via Whole-Body [¹⁸F]FDG-PET/CT: Pancreas Size as a Biomarker for Cachexia in Head and Neck Squamous Cell Carcinoma." (L2)

  1. In table 1, can the author include the available Pancreatic volume data?

We appreciate this suggestion and agree that including pancreatic volume data strengthens the manuscript. The mean pancreatic volumes for the respective groups have been added to result section, and the reference to Supplementary Table 1 has been highlighted again for further details (L245).

  1. Fig5B include Y axis label.

Thank you for pointing this out. We have implemented your suggestion and added the Y-axis label to Figure 5B (L317).

  1. Although it can be indicated that “Pancreatic volume” can be a predictor of cachexia but it is not confirmed. How will the authors clarify that?

We acknowledge that, while pancreatic volume can predict cachexia in our patient group, further validation is needed to confirm the value of this measurement as a clinically relevant biomarker. In the discussion section, we have provided a more detailed exploration of the potential mechanisms linking pancreatic volume to cachexia (L390). However, we recognize that fully confirming pancreatic volume as a biomarker is beyond the scope of the current study. To further address this, we have initiated a prospective study in lung cancer patients that leverages machine learning-based imaging analysis and evaluates pancreatic enzymes (amylase, lipase) and hormones related to energy metabolism (insulin, glucagon). Cortisol, due to its role in anabolic and catabolic functions, will also be assessed. This future data will be essential before advancing towards multi-center studies to validate pancreatic volume as a biomarker for cachexia (NCT05912465).

  1. Some of the statements are not “totally” correct. Eg. “This malignancy, primarily managed by Ear, Nose, and Throat (ENT) specialists, affects critical areas such as the oral cavity, pharynx, and larynx—regions essential for nutritional intake.” There is a significant contribution from the oncologists, radiation dosimetrists etc…so the authors should edit that sentence. Moreover, I could not see such thing in their cited ref #2

We apologise for our oversimplification and we have revised the text to better reflect the interdisciplinary nature of disease management. Additionally, we have reviewed and verified the accuracy of the reference. The revised paragraph can be found at L50.

  1. Again, while reporting the distribution the authors have cited a reference dating back to 2018. They should cite recent papers.

Thank you for your comments regarding some dated references. We have reviewed carefully all citations >5 years old and replaced some references, such as Kotler, D.P. (2000) and Klein, G.L. et al. (2013), with more current studies. However, we chose to retain some highly impactful studies that continue to be relevant in the field of cachexia research. We hope this approach is agreeable to the reviewer and these changes are reflected in the revised manuscript.

Reviewer 2 Report

Comments and Suggestions for Authors

The present manuscript provides original retrospectively compiled patient data 

with a putative relevance for the solution of a major problems in the treatment of head and neck cancer, namely iatrogenic (radiotherapy, chemotherapy)-induced toxicity. The approach is critical, and limitations of the study are duly considered.

The observation about pancreatic volume is thought-provoking.

The following notes might help to improve the manuscript.

One.  All abbreviations should be explained when first used. For example WLGS (L34), explain IMAT, SAT, SKM, VAT (L238), etc.

Two. Figures need labeling and more extensive legends so as to be readable without the text. In Figure 2 A and B, PET positive structures should be specified on behalf of the non-expert. Figure 2C is confusing; colors need to be explained. 

In Figure 5 A, numerical values in x-axis need explanation.

What is the meaning of the colors in Figure 5B?

Three. L333 – 340. Could the authors provide a more concrete image of inter-organ metabolic relationship (metabolic connectivity)? Are molecular signals known?  L360. What is “chronic inflammation  independent of blood CRP levels”? 

Four. L447. The authors make the interesting suggestion that “A reliable and early diagnosis of cachexia with such a tool could support the clinical development  of novel therapeutic interventions for this devastating condition”. Could they make a suggestion about the kind of therapeutic interventions and clinical trials?

Author Response

The present manuscript provides original retrospectively compiled patient data 

with a putative relevance for the solution of a major problems in the treatment of head and neck cancer, namely iatrogenic (radiotherapy, chemotherapy)-induced toxicity. The approach is critical, and limitations of the study are duly considered.

The observation about pancreatic volume is thought-provoking.

The following notes might help to improve the manuscript.

  1. All abbreviations should be explained when first used. For example WLGS (L34), explain IMAT, SAT, SKM, VAT (L238), etc.

Thank you for your input. We have reviewed the use of abbreviations and implemented the explanation of the abbreviations in the manuscript, whenever first used (L32, L245)

2a. Figures need labeling and more extensive legends so as to be readable without the text. In Figure 2 A and B, PET positive structures should be specified on behalf of the non-expert. Figure 2C is confusing; colors need to be explained. 

Thank you for the helpful feedback and we hope the revised figures are improved in their clarity. In Figures 2A and 2B, PET-positive structures are now more clearly labelled for non-expert readers. In Figure 2C, we have clarified the meaning of the colors used for organ segmentation. The figure legends have also been expanded with relevant details to ensure the figures are understandable without referring to the main text.

2b. In Figure 5 A, numerical values in x-axis need explanation. What is the meaning of the colors in Figure 5B?

Thank you for the comment about the figure containing SHAP interpretation, which can be complex. We have adjusted the x-axis in Figure 5A to be more intuitive, clearly highlighting the trend towards the ML endpoint of the cachexia phenotype. In Figure 5B, we have replaced the colors with distinct shades of grey for better visual distinction, while maintaining the focus on the AUC as the key metric being conveyed (L317)

3a. L333 – 340. Could the authors provide a more concrete image of inter-organ metabolic relationship (metabolic connectivity)? Are molecular signals known?  

Thank you for your question. Recent studies have highlighted key molecular signals such as mTOR, AMPK, and hormonal regulators like insulin and glucagon, which play a role in inter-organ metabolic communication during cachexia. While our study does not directly evaluate these mechanisms, we discuss their relevance in the context of increased metabolic connectivity in cachectic patients, reflecting the complex metabolic reorganization characteristic of this condition (L369).

3b. L360. What is “chronic inflammation independent of blood CRP levels”? 

Thank you for the question. In this context, "chronic inflammation independent of blood CRP levels" refers to the potential detection of inflammation through [¹⁸F]FDG uptake in tissues, even when traditional blood markers such as CRP or IL-6 are unavailable or not elevated. [¹⁸F]FDG PET/CT can identify metabolic activity in inflamed tissues, such as adipose tissue, which may signal an inflammatory state that blood markers alone might not detect. In our study, this was reflected in the elevated [¹⁸F]FDG uptake in subcutaneous fat, suggesting underlying inflammation despite the lack of available CRP or IL-6 data in the cohort. We have revised the manuscript to clarify this point, as noted in the relevant paragraph (L384).

  1. L447. The authors make the interesting suggestion that “A reliable and early diagnosis of cachexia with such a tool could support the clinical development of novel therapeutic interventions for this devastating condition”. Could they make a suggestion about the kind of therapeutic interventions and clinical trials?

Thank you for your request for clarification. We have addressed this point in the discussion. Current treatments, such as nutritional support and cortisol therapy, are often ineffective due to late diagnosis. Early detection through tools like [¹⁸F]FDG-PET/CT could improve treatment by enabling earlier interventions. Novel therapeutic options, including GDF-15 antibodies, SARMs, anamorelin, and anti-myostatin therapies, were also mentioned as potential approaches to address muscle wasting and improve outcomes (L480).

Round 2

Reviewer 1 Report

Comments and Suggestions for Authors

The authors have clarified most of my questions. I have two suggestions:

1. Biomarker is strong word with lot of biological expectations which the authors could not provide and validate so its best to edit the title as, "Systemic Metabolic and Volumetric Assessment via Whole Body [¹⁸F]FDG-PET/CT: Pancreas Size predicts Cachexia in Head and Neck Squamous Cell Carcinoma?"

2. In Figure 4 nomogram (top panel) some of the numerals particularly for tumor volume, SKM SUV, Pancreas SUV are not clear due to overlapping of X-axis. Edit it so that its more clearly visible.

Author Response

We are grateful for the prompt review and the insightful suggestions, which we believe significantly enhance the manuscript.

1. Biomarker is strong word with lot of biological expectations which the authors could not provide and validate so its best to edit the title as, "Systemic Metabolic and Volumetric Assessment via Whole Body [¹⁸F]FDG-PET/CT: Pancreas Size predicts Cachexia in Head and Neck Squamous Cell Carcinoma?"

We appreciate the reviewer’s input. We had also considered this title and are content that it more accurately reflects the study’s focus. Therefore, we have updated the title to: "Systemic Metabolic and Volumetric Assessment via Whole Body [¹⁸F]FDG-PET/CT: Pancreas Size Predicts Cachexia in Head and Neck Squamous Cell Carcinoma?".

2. In Figure 4 nomogram (top panel) some of the numerals particularly for tumor volume, SKM SUV, Pancreas SUV are not clear due to overlapping of X-axis. Edit it so that its more clearly visible.

We apologize for the layout issue caused by the incorrect conversion of the SVG format into the PDF. We have addressed this by using a PNG version of Figure 4 in the revised manuscript. We hope this resolves the issue and ensures the figure’s clarity for readers.
